# Atrial Pacing Affects Left Atrial Morphological and Functional Parameters Early after Pacemaker Implantation

**DOI:** 10.3390/medicina58091283

**Published:** 2022-09-15

**Authors:** Mindaugas Viezelis, Gintare Neverauskaite-Piliponiene, Agne Marcinkeviciene, Eligija Teleisyte, Tomas Kazakevicius, Vytautas Zabiela, Vilius Kviesulaitis, Renaldas Jurkevicius, Aras Puodziukynas

**Affiliations:** 1Department of Cardiology, Lithuanian University of Health Sciences, A. Mickeviciaus Str. 7, LT-44307 Kaunas, Lithuania; 2Department of Cardiology, Hospital of Lithuanian University of Health Sciences Kaunas Clinics, Eiveniu Str. 2, LT-50161 Kaunas, Lithuania

**Keywords:** left atrial function, right atrial pacing, left atrial strain, pacemaker, cardiac pacing

## Abstract

*Background and Objectives*: Chronic atrial stimulation might impair left atrial (LA) function. We aimed to assess the impact of atrial pacing burden on LA volumetric and functional parameters after implantation of a dual-chamber pacemaker. *Materials and Methods*: The prospective study included 121 subjects with sinus node disease (SND) or atrioventricular block (AVB) that received a dual-chamber pacemaker. After device implantation, the subjects were divided into three groups: (1) SND with a lower rate (LR) set to 60 bpm and rate response enabled; (2) AVB with an LR set to 60 bpm and no rate response; and (3) AVB with an LR set to 40 bpm and no rate response. Subjects were followed at one and three months. Two subsets of patients with high and low atrial pacing burdens accompanied by low ventricular pacing percentage were analyzed. LA function was evaluated with volumetric and strain parameters from transthoracic echocardiography. *Results*: The high atrial low ventricular pacing group consisted of 38 subjects, and the low atrial low ventricular group consisted of 22 subjects. When looking at the change in volumetric parameters, we observed a tendency for volumes to increase in both groups; however, only minimal LA volume reached statistical significance at three months in the high atrial pacing group. A trend towards the lowering of an active emptying fraction at one month (*p* = 0.076) became significant at three months (*p* = 0.043), and was also only observed in the same group. Moreover, a decrease in the tendency to reach statistical significance at three months for reservoir and contractile strain parameters and stiffness index was only observed in the high atrial pacing group. Meanwhile, in the low atrial pacing burden group, all parameters remained significantly unchanged. *Conclusions*: The burden of right atrial pacing in patients with preserved left ventricular function negatively influences functional and morphologic LA parameters. Moreover, negative effects take place soon after pacemaker implantation and appear to be sustained.

## 1. Introduction

Dual-chamber cardiac pacing improves clinical outcomes and quality of life for patients with bradyarrhythmia. Atrioventricular block (AVB) and sinus node dysfunction (SND) are two of the main indications for cardiac pacing [1,2]. However, despite the obvious benefits of pacemaker implantation, there is an unfavorable side effect of it. In 2003, the CTOPP study showed that dual-chamber pacing (DDD mode) is superior to ventricular pacing (VVI mode) for the development of atrial fibrillation (AF). It is more physiological [3]. Ventricular pacing leads to a remodeling of the heart and may increase the risk of heart failure (HF), AF, and death [4,5]. It is well established that right ventricular (RV) pacing greater than 40% induces interventricular and intraventricular desynchrony and impairs left ventricular (LV) function [6,7]. Apical RV pacing has a detrimental effect on left atrial (LA) function [8]. However, atrial pacing burden has not been evaluated in the study. Nevertheless, chronic atrial stimulation might impair LA function and lead to arrhythmias because of its mechanical and neuro endocrine functions [9,10,11].

Martens et al. recently presented a study stating that atrial pacing in CRT patients has a negative effect on LA morphology, has a function that leads to a higher risk of AF onset, and worsens HF [12]. However, to date data on the downside of chronic atrial pacing and optimal settings is lacking in patients with preserved LV ejection fraction (EF). It is not known how early these changes occur after pacemaker implantation. By evaluating atrial remodeling, an optimal atrial pacing strategy could be determined for specific patient groups.

We aimed to assess the impact of atrial pacing burden on LA volumetric and functional parameters after implantation of a dual-chamber pacemaker.

## 2. Materials and Methods

### 2.1. Study Population

The prospective study was conducted at Lithuanian University of Health Sciences, Department of Cardiology from June 2020 to November 2021. The study in a prospective manner enrolled 121 subjects who received dual-chamber pacemaker implantation because of documented SND or advanced AVB. Subjects underwent standard device implantation procedure with the RA lead primarily positioned into the RA appendage and the RV septal judged by fluoroscopy. The exclusion criteria were history of persistent AF, significant structural heart disease, left ventricular ejection fraction (LVEF) <50%, and inability to comply with the follow-up regime. Subjects after device implantation were divided into three groups: (1) SND with a lower rate (LR) set to 60 bpm and rate response enabled; (2) AVB with a LR set to 60 bpm and no rate response; and (3) AVB with a LR set to 40 bpm and no rate response. Patients were followed up at one- and three-months visits. During enrollment, clinical characteristics (gender, age, indication, and previous medical history) were collected. The study was approved by the National Ethical Committee and written informed consent was obtained from all patients.

### 2.2. Echocardiography

Conventional two-dimensional transthoracic echocardiography and color TDI were performed the next day after pacemaker implantation. A commercially available ultrasound system (model EPIQ 7, Philips Medical Systems, Andover, MA, USA) was used. All investigations were performed by experienced echocardiographers.

High-quality apical 4-chamber, 2-chamber, 3-chamber, and parasternal views over 4 consecutive cardiac cycles were recorded and stored for further off-line analysis. Sector width was optimized to maintain high frame rate, and LA foreshortening was avoided. All echocardiographic analysis was performed by the same reader blinded to other study-related parameters including atrial and ventricular pacing burden.

### 2.3. Left Atrium Evaluation

LA volumes were measured with two-dimensional echocardiographic images by area–length method from apical 4-chamber and 2-chamber views as recommended by the American Society of Echocardiography [13]. Endocardial tracing excluded the atrial appendage and pulmonary veins. LA length was measured from mitral valve level to the back wall. The maximal length and area of LA for volume calculation (LAVmax) was measured in end-systole and the pre-atrial contraction volume (LAVpre)—at the start of P wave on ECG and the minimum volume (LAVmin)—at the end-diastole assessed on ECG. LA volumetric parameters were indexed to body surface area. LA emptying fractions were calculated as follows: (1) LA passive emptying fraction, defined as (LAVmax − LAVpre)/LAVmax × 100%; (2) LA active emptying fraction, defined as (LAVpre − LAVmin)/LAVpre × 100%; and (3) LA total emptying fraction, defined as (LAVmax − LAVmin)/LAVmax × 100%. For the analysis of echocardiographic parameters, dedicated software (EchoPAC PC SWO version 112.x.x, General Electric, Horten, Norway) was used.

Strain parameters were analyzed using a commercially available software package (Philips QLAB version 15.0 Philips Medical Systems, Andover, MA, USA). An atrial cycle was used as the reference point [14]. To generate LA strain curves, the LA endocardial border was automatically traced in the apical 4-chamber and 2-chamber views. The tracing was adjusted by the reader if necessary and approved. LA stiffness index was calculated as the ratio of E/e′ to LA reservoir strain as previously defined [15].

Mitral inflow pattern was evaluated by E and A ratio. Septal and lateral mitral annular velocities (E′) were obtained by pulsed-wave tissue Doppler imaging and were averaged. LV diastolic filling pressure index was measured by the ratio of transmitral early diastolic velocity to the mitral averaged annulus velocity by pulse TDI (E/e′). The value of >15 signifies elevated LV filling pressure [16].

### 2.4. Follow-Up Visit Procedure

Subjects were followed up in the same implanting center. At one and three months, follow-up echocardiography was performed. A 12-lead ECG was taken for every patient to precisely assess the underlying rhythm. A pacemaker read-out was done taking note of atrial and ventricular pacing burdens.

A cutoff threshold above and below 40% of right atrial and ventricular pacing was chosen. To investigate the right atrial pacing effect on LA functional parameters while minimizing RV pacing effect, we looked at the groups with high and low RA pacing burden that maintained low RV pacing. Grouping into subgroups based on RA and RV pacing burden did not change during the follow-up period.

### 2.5. Statistical Analysis

Data are presented as mean ± SD for continuous variables or as percentages for categorical variables, unless otherwise indicated. The Shapiro–Wilk test was used to determine the distribution of the data. Continuous variables were compared using Student’s *t*-test or Mann–Whitney U test as appropriate. Continuous variables in the same group of subjects at the follow-up visits were compared by paired samples with the Student *t*-test or Wilcoxon test. Categorical variables were compared using the Chi-square test. To assess baseline LA morphometric parameters distribution at baseline between all groups, an ANOVA model was used. Statistical significance was chosen to be *p*-value < 0.05. Intraobserver variabilities for speckle tracking measurements were tested in 10 randomly selected patients using the identical cine-loops for each view calculating intraclass correlation. Statistical analysis was performed using SPSS Statistics for Macintosh, Version 24.0 (IBM Corp., released 2016, Armonk, NY, USA). 

## 3. Results

### 3.1. Baseline Parameters

A total of 121 subjects (mean age 74.5 ± 10.4 years; 47 (42.7%) men) were enrolled in the study. Intraclass correlation coefficients for intraobserver variability were 0.962, 0.903, and 0.912 for reservoir, conduit, and contractile strains, respectively. The baseline clinical characteristics are summarized in Table 1. The AVB 40 bpm subgroup was numerically slightly younger, had higher body surface area, and had higher body mass index, though not reaching statistical significance. In the same group we observed not significantly higher LA volumetric parameters. There were less subjects diagnosed with diabetes mellitus in the SND group. Baseline medication distribution and strain values did not differ between groups (Table 1).

### 3.2. Pacing Distribution

Because of COVID-19-related travel and working restrictions, ten subjects did not complete the follow-up period, and two subjects developed persistent atrial fibrillation. In the remaining 109 subjects, echocardiography and pacemaker read-outs were performed. The mean RA and RV cumulative pacing percentage are presented in Figure 1. The pacing percentage remained significantly different during the follow-up period between the subgroups regarding both RA and RV pacing burden (*p* < 0.001). Pacing burden remained unchanged in each of the subgroups during the study period (*p* > 0.05). The high RA low RV pacing group consisted of 38 and the low RA low RV pacing consisted of 22 subjects (Figure 2).

### 3.3. Comparison of High and Low Right Atrial Pacing with Low Right Ventricular Pacing Groups

The baseline parameters between the high and low RA pacing groups are shown in Table 2 and Table 3. There were more subjects with diabetes mellitus in the low RA pacing group. LA volumetric and functional parameters were not different at baseline.

When looking at change of volumetric parameters, we observed a tendency for volumes to increase in both groups (Table 4 and Table 5), however only LAVmin reached statistical significance at three months in the high RA pacing group. A trend towards lowering of active emptying fraction at one month (*p* = 0.076) that became significant at three months (*p* = 0.043) was also only observed in the same group. Again, a tendency to decrease reaching statistical significance at three months of reservoir and contractile strain parameters and stiffness index was only observed in the high RA pacing group. Meanwhile in the low RA pacing burden group, all parameters remained significantly unchanged.

## 4. Discussion

To our knowledge, this is the first study investigating atrial pacing burden effect in patients with preserved LVEF undergoing dual chamber pacemaker implantation and addressing RV pacing burden. It adds important novel information about atrial pacing effect on LA function after conventional dual chamber pacemaker implantation, which remains the most common mode of pacing. The main findings are the following: (1) higher atrial pacing burden was associated with a negative effect on LA remodeling parameters following dual chamber pacemaker implantation; (2) higher atrial pacing negatively affected systolic and diastolic LA function; and (3) the negative effects of RA pacing appear soon after pacemaker implantation and appear to be sustained.

It has been previously demonstrated the RV pacing, besides affecting LV function, can also have detrimental effects on LA function [6]. However, it must be acknowledged that in the setting of dual chamber pacemakers there is another relevant factor involved that has less emphasis put on right atrial pacing. However, data evaluating these effects are limited.

LA size is a predictor of the development of new HF. When HF is present, LA enlargement and dysfunction are important predictors of clinical outcomes. Enlargement of the left atrium is also a marker of disease severity and predicts adverse cardiovascular events in heart failure with preserved ejection fraction (HFpEF). In a study that included 89 normal subjects, 38 asymptomatic hypertensive patients, and 183 patients with HFpEF, NYHA class, diastolic blood pressure, age, and LA dimensions were independent predictors of mortality [17]. In our study we observed a tendency for LA volumes to increase in both investigated groups, though it only reached statistical significance in the high RA pacing group. However, it has to be noted that the absolute change was not different between the two groups, and a longer observational period is needed to observe if the pattern is maintained.

A study by Liang et al. has previously investigated the acute hemodynamic effect of atrial pacing vs. sensing in patients with heart failure with reduced ejection fraction (HFrEF) undergoing CRT implantation. Their study showed that pacing mode is associated with intra-atrial dyssynchrony and might limit LA preload contribution of LV stroke volume [18]. However, it was unclear if chronic RA pacing and its burden affects LA structure and function. Additionally, this study used tissue Doppler imaging analysis which is prone to technical issues. What is more, choosing sensed or paced mode might lead to different heart rates, thus affecting hemodynamics and analysis. Martens et al. investigated RA pacing burden effect on interatrial desynchrony in a similar patient subset (HFrEF undergoing CRT implantation) [12]. Their study showed that higher RA pacing burden was linked to diminished LA reverse remodeling response and the worsening of atrial function and structure. They noticed atrial pacing’s detrimental effect on LA reservoir and contractile functions assessed by strain analysis. In our study we observed that significant LA functional changes occurred in only high RA pacing groups, evaluating both volume derived functional parameters and strain parameters. What is more, we observed a tendency to decrease in strain parameters representing all LA functions: reservoir, conduit, and contractile. This finding might be explained by pronounced intra-atrial delay and deterioration in strain parameters while increasing in volumetric ones. Martens et al. propose that conduit function was not affected, since it mostly depends on LV suction/filling. However, it must be noted that the study was done after CRT implantation when it is expected to have improvement in LV function. We speculate that such improvement could have counterbalanced change in conduit function. Since our study included patients with preserved LVEF, we observed change in all three strain parameters. It has been previously shown than large LA volume is associated with interatrial conduction delay [19]. In our study both analyzed groups had increased baseline LA volumes. High RA pacing could have promoted interatrial conduction delay, thus leading to less synchronous atria contraction and less than normal interatrial septal compliance. Moreover, if the delay was significant enough, it could have led to mistimed LA contraction against mitral valve, further increasing LA pressure and affecting its function.

It must be noted that previous studies explored pacing effects on LA function and morphology in CRT setting. Though CRT is more physiological than only RV pacing, it is still artificial. What is more, when choosing CRT as a means of treatment, one tries to achieve the highest ventricular pacing percentage possible. In our study we analyzed patients with low ventricular pacing. Furthermore, having as a baseline HFrEF subjects, often accompanied by secondary mitral regurgitation, a higher degree of LA enlargement experience from such studies might not directly translate to subjects with normal LVEF and no significant structural heart disease. What is more, ventricular pacing in our study was not negligible. The ventricular pacing site also plays a part in LA function and affects its remodeling. Apical pacing has been associated with AF, HF, and mortality [19]. It has been shown that septal pacing is associated with more physiological LV electromechanical activation and relaxation and consequently better LA function [20,21]. Though in our study RV position during implantation has been based by fluoroscopy and judged as septal, it has been shown that evaluation without cardiac computed tomography guidance is imprecise [22].

In the study conducted by Sade et al. looking into AF onset or recurrence in CRT patient population, a higher RA pacing burden predicted AF-recurrence or new-onset AF, respectively [23]. Maintenance of LA function was associated with freedom from AF. A similar effect was observed in another study where AF onset or recurrence was linked to worsening or lack of improvement in LA function after CRT implantation [12]. Adelstein et al. concluded in the study following patients after CRT implantation that compared to atrial sensing, atrial pacing is associated with a two-fold increased risk of post-CRT AF [24]. We believe our study duration is too short to show outcomes regarding AF. However, LA function was maintained only in the group of low RA pacing thus possibly placing subjects in the other group at higher risk of AF development of relapse.

Another novel index that has been previously shown to have a relationship to HF development is the LA stiffness index [15]. In the study, the LA stiffness index, though remaining significantly lower than in patients with diastolic or systolic heart failure, in patients with diastolic dysfunction it was higher than in controls. In our study we have observed significant change in the LA stiffness index only in the high atrial pacing group, thus suggesting a diastolic LA dysfunction occurring with higher atrial pacing burden.

Though our findings are interesting from a pathophysiologic point of view, an equally important clinical practice aspect is important. As with the well-established detrimental effect of right ventricular pacing on LV function and further device programming considerations evident from previous studies, transferring experience to the level of the atria seems logical, as almost no such data are available [7]. RA pacing can be considered as a modifiable factor. What is more, atrial pacing recommendations in the current pacing guidelines regarding rate selection in different patient subsets are lacking [1,25]. An example would be, when feasible, promoting atrial sensing by programming lower base rate or hysteresis. We cannot draw a conclusion about the precise atrial pacing burden that is critical to affect LA function, though our chosen 40% pacing burden limit appears to perform well. In case of expected high atrial pacing burden, an alternative pacing site instead of the right atrial appendage may be chosen, thus facilitating conduction through fibers interconnecting the RA and LA (Bachmann’s bundle), or at the coronary sinus ostium might limit inter- and intra-atrial desynchrony described in previous studies [12,18,26].

### Study Limitations

Firstly, we must acknowledge a relatively small sample size in our study. Second, strain analysis has an image quality-dependent modality. However, we report good intraobserver variability. Thirdly, though pacing programming has been strictly maintained regarding lower rate and rate response during study period, other programming parameters (i.e., AV delay) might have been biased by the implanter. Fourth, a relatively short follow-up period does not allow drawing conclusions on arrhythmia occurrence and further LA function development. Fifth, ventricular pacing burden of up to 40% was considered low RV pacing though it was not negligible. Sixth, the right ventricular pacing site can also have an effect on the left atrium, although it has not been considered in our study.

## 5. Conclusions

The burden of RA pacing in patients with preserved LVEF negatively influences functional and morphologic LA parameters. Moreover, negative effects take place soon after pacemaker implantation and appear to be sustained.

## Figures and Tables

**Figure 1 medicina-58-01283-f001:**
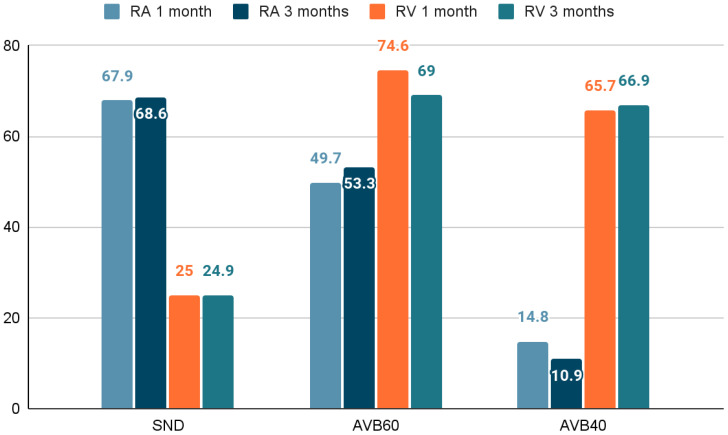
Pacing percentage. AVB40-atrioventricular block group with a lower rate set to 40 bpm and no rate response. AVB60-atrioventricular block group with a lower rate set to 60 bpm and no rate response. SND—Sinus node disease group with a lower rate set to 60 bpm and rate response enabled. RA—right atrium; RV—right ventricle.

**Figure 2 medicina-58-01283-f002:**
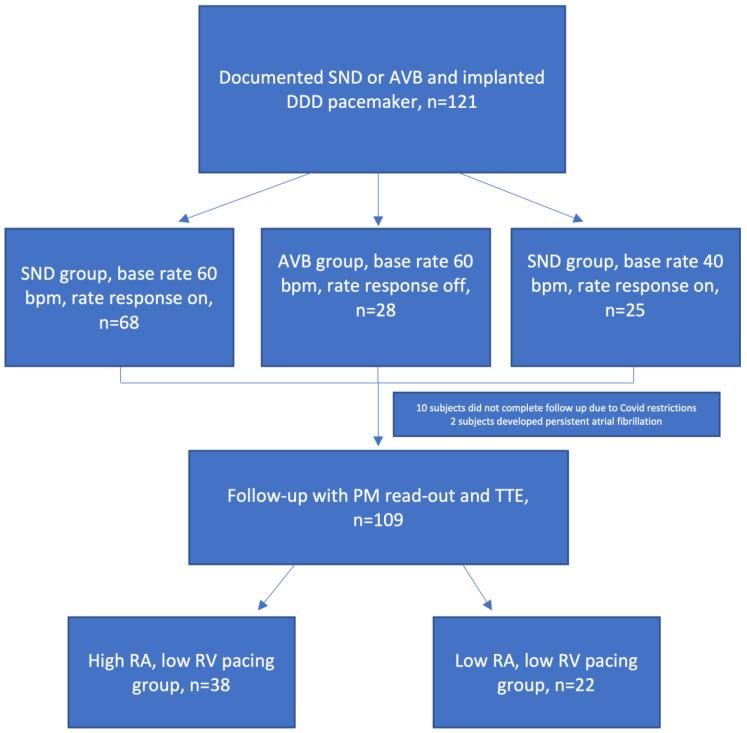
Flowchart of study subjects. AVB—atrioventricular block; bpm—beats per minute; PM-pacemaker; RA—right atrial, RV—right ventricular; SND—sinus node disease; TTE—transthoracic echocardiography.

**Table 1 medicina-58-01283-t001:** Baseline parameters based on indication and base rate.

	SND Group ^a^ (*n* = 68)	AVB60 Group ^b^ (*n* = 28)	AVB40 Group ^c^ (*n* = 25)	*p*-Value
Age, years	75.1 ± 9.5	75.9 ± 11.3	69.3 ± 103	0.104
Male, *n* (%)	23 (33.8)	14 (41.2)	10 (52.6)	0.332
Body surface area	1.91 ± 0.21	1.90 ± 0.25	1.98 ± 0.25	0.472
Body mass index, kg/m^2^	28.9 ± 4.6	28.7 ± 6.6	30.5 ± 5.0	0.494
Medical history				
Hypertension, *n* (%)	63 (92.6)	26 (92.3)	25 (100)	0.380
Diabetes mellitus, *n* (%)	5 (7.4)	8 (28.6)	6 (24.0)	**0.015**
Paroxysmal atrial fibrillation, *n* (%)	27 (39.7)	6 (21.4)	11 (44.0)	0.161
Coronary artery disease, *n* (%)	27 (39.7)	9 (32.1)	10 (40.0)	0.766
Chronic renal failure, *n* (%)	3 (4.4)	3 (10.7)	3 (12.0)	0.350
Medications				
ACE inhibitors/ARB	64 (94.1)	27 (96.4)	22 (88.0)	0.438
BAB	51 (75.0)	21 (75.0)	22 (88.0)	0.380
MRA	18 (26.5)	7 (25.0)	5 (20.0)	0.814
Other diuretic	33 (48.5)	14 (50.0)	13 (52.0)	0.956
Non dihydropyridine CCB	1 (1.5)	0 (0)	0 (0)	0.675
Statin	28 (35.3)	10 (35.6)	10 (40.0)	0.883
LA volumetric parameters				
LAVmax, mL	74.6 ± 20.5	71.6 ± 22.8	83.0 ± 28.4	0.359
LAVmax index, mL/m^2^	38.9 ± 9.3	37.4 ± 9.5	40.6 ± 12.8	0.712
LAVpre, mL	54.2 ± 17.0	49.7 ± 17.7	57.4 ± 17.7	0.382
LAVpre index, mL/m^2^	28.3 ± 8.0	25.9 ± 7.5	28.1 ± 7.9	0.544
LAVmin, mL	39.2 ± 14.0	37.4 ± 14.9	43.2 ± 15.6	0.379
LAVmin index, mL/m^2^	20.5 ± 6.8	19.3 ± 6.3	21.1 ± 7.2	0.682
LV parameters				
LVEDD, mm	49.6 ± 4.7	49.7 ± 5.5	50.9 ± 4.9	0.655
LVEDD index, mL/m^2^	26.1 ± 2.4	26.3 ± 2.6	25.2 ± 2.3	0.302
LVEF, %	58.4 ± 4.8	57.8 ± 4.7	59.2 ± 5.0	0.808
E/A	0.88 ± 0.40	0.80 ± 0.41	0.87 ± 0.30	0.983
E/e′	8.7 ± 3.0	9.5 ± 4.6	11.1 ± 3.9	0.062
LA functional parameters				
Total emptying fraction, %	48.1 ± 8.4	48.9 ± 8.7	47.8 ± 7.6	0.894
Passive emptying fraction, %	27.6 ± 8.4	31.0 ± 9.6	29.5 ± 7.8	0.200
Active emptying fraction, %	28.2 ± 9.4	25.7 ± 8.6	25.9 ± 8.1	0.409
LA strain parameters				
Reservoir strain, %	23.5 ± 10.1	21.1 ± 7.3	22.1 ± 12.7	0.521
Conduit strain, %	−10.7 ± 4.9	−9.2 ± 6.5	−11.0 ± 6.1	0.144
Contractile strain, %	−12.8 ± 8.5	−11.9 ± 6.9	−11.1 ± 10.4	0.513
Stiffness index	0.50 ± 0.51	0.51 ± 0.32	0.78 ± 0.69	0.184

ACE—angiotensin converting enzyme; ARB—angiotensin receptor blockers; AVB—atrioventricular block; BAB—beta-adrenoceptor blockers; CCB—calcium channel blocker; LA—left atrium; LAVmax—maximal left atrium volume; LAVmin—minimal left atrium volume; LAVpre—pre-atrial contraction left atrium volume; LV—left ventricle; LVEDD—left ventricle end diastolic diameter; LVEF—left ventricular ejection fraction; MRA—mineralocorticoid receptor antagonists; SND—sinus node disease. ^a^ Sinus node disease group with a lower rate set to 60 bpm and rate response enabled. ^b^ Atrioventricular block group with a lower rate set to 60 bpm and no rate response. ^c^ Atrioventricular block group with a lower rate set to 40 bpm and no rate response.

**Table 2 medicina-58-01283-t002:** Baseline parameters based on high and low atrial pacing, low right ventricular pacing.

	High RA Low RV Group (*n* = 38)	Low RA Low RV Group (*n* = 22)	*p*-Value
Age, years	73.6 ± 10.2	69.0 ± 13.3	0.349
Male, *n* (%)	12 (31.7)	8 (36.4)	0.705
Body surface area	1.89 ± 0.2	2.0 ± 0.29	0.404
Body mass index, kg/m^2^	28.4 ± 4.2	30.4 ± 6.4	0.312
Medical history			
Hypertension, *n* (%)	35 (92.1)	19 (86.4)	0.475
Diabetes mellitus, *n* (%)	3 (7.9)	6 (27.2)	**0.043**
Paroxysmal atrial fibrillation, *n* (%)	15 (39.4)	7 (31.8)	0.553
Coronary artery disease, *n* (%)	14 (36.8)	6 (27.3)	0.449
Chronic renal failure, *n* (%)	2 (5.2)	3 (13.6)	0.258
Medications			
ACE inhibitors/ARB	36 (94.7)	2 (90.1)	0.567
BAB	29 (76.3)	16 (72.7)	0.757
MRA	10 (26.3)	5 (22.7)	0.757
Other diuretic	17 (44.7)	11 (50.0)	0.694
Non dihydropyridine CCB	1 (2.6)	0 (0)	0.443
Statin	12 (31.6)	8 (36.4)	0.705
LV parameters			
LVEDD, mm	49.2 ± 4.5	49.6 ± 5.3	0.600
LVEDD index, mL/m^2^	26.2 ± 2.2	25.4 ± 2.2	0.358
LV EF, %	58.4 ± 4.9	58.2 ± 5.0	0.798
E/A	0.89 ± 0.37	0.84 ± 3.4	0.406
E/e′	8.4 ± 2.6	8.7 ± 3.7	0.842

ACE—angiotensin converting enzyme; ARB—angiotensin receptor blockers; AVB—atrioventricular block; BAB—beta-adrenoceptor blockers; CCB—calcium channel blocker; LA—left atrium; LAVmax—maximal left atrium volume; LAVmin—minimal left atrium volume; LAVpre—pre-atrial contraction left atrium volume; LV—left ventricle; LVEDD—left ventricle end diastolic diameter; LVEF—left ventricular ejection fraction; MRA—mineralocorticoid receptor antagonists; RA—right atrium, RV—right ventricle, SND—sinus node disease.

**Table 3 medicina-58-01283-t003:** Baseline left atrial parameters between high and low right atrial, low right ventricular pacing groups.

	High RA Low RV Group	Low RA Low RV Group	*p*-Value
LA volumetric parameters			
LAVmax, mL	73.2 ± 17.3	72.5 ± 27.2	0.916
LAVmax index, mL/m^2^	38.7 ± 7.9	36.5 ± 11.9	0.433
LAVpre, mL	53.7 ± 14.3	50.2 ± 20.2	0.479
LAVpre index, mL/m^2^	28.3 ± 6.8	25.2 ± 8.8	0.164
LAVmin, mL	38.2 ± 11.8	35.6 ± 16.1	0.511
LAVmin index, mL/m^2^	20.2 ± 5.9	17.9 ± 7.3	0.230
LA functional parameters			
Total emptying fraction, %	48.1 ± 8.2	51.6 ± 7.2	0.979
Passive emptying fraction, %	26.7 ± 8.2	31.1 ± 7.6	0.586
Active emptying fraction, %	29.2 ± 8.7	29.7 ± 8.6	0.903
LA strain parameters			
Reservoir strain, %	25.9 ± 10.3	23.7 ± 9.5	0.493
Conduit strain, %	−11.9 ± 5.3	−11.0 ± 3.7	0.565
Contractile strain, %	−14.0 ± 9.0	−12.7 ± 7.6	0.633
Stiffness index	0.41 ± 0.27	0.45 ± 0.32	0.639

LA—left atrium; LAVmax—maximal left atrium volume; LAVmin—minimal left atrium volume, LAVpre—pre-atrial contraction left atrium volume; RA—right atrium, RV—right ventricle.

**Table 4 medicina-58-01283-t004:** High right atrial low right ventricular pacing group.

	Baseline	1 Month	3 Months	*p*-Value Baseline vs. 1 Month	*p*-Value Baseline vs. 3 Months
LA volumetric parameters					
LAVmax, mL	73.2 ± 17.3	77.8 ± 21.1	75.8 ± 20.1	0.442	0.367
LAVmax index, mL/m^2^	38.7 ± 7.9	41.0 ± 10.4	40.1 ± 10.0	0.424	0.376
LAVpre, mL	53.7 ± 14.3	55.5 ± 16.2	57.3 ± 17.5	0.294	0.161
LAVpre index, mL/m^2^	28.3 ± 6.8	29.3 ± 8.2	30.2 ± 8.4	0.261	0.186
LAVmin, ml	38.2 ± 11.8	41.3 ± 14.6	42.7 ± 13.7	0.169	**0.038**
LAVmin index, mL/m^2^	20.2 ± 5.9	21.7 ± 7.3	22.6 ± 7.5	0.190	**0.039**
LA functional parameters					
Total emptying fraction, %	48.1 ± 8.2	47.6 ± 8.6	44.9 ± 9.8	0.678	**0.033**
Passive emptying fraction, %	26.7 ± 8.2	28.6 ± 9.4	24.5 ± 9.7	0.398	0.401
Active emptying fraction, %	29.2 ± 8.7	26.5 ± 8.5	25.7 ± 8.9	0.076	**0.043**
LA strain parameters					
Reservoir strain, %	25.9 ± 10.3	24.4 ± 9.5	21.1 ± 9.9	0.315	**0.003**
Conduit strain, %	−11.9 ± 5.3	−11.8 ± 6.4	−10.0 ± 5.3	0.798	0.086
Contractile strain, %	−14.0 ± 9.0	−12.7 ± 7.0	−11.1 ± 7.8	0.342	**0.018**
Stiffness index	0.41 ± 0.27	0.46 ± 0.33	0.67 ± 0.65	0.478	**0.001**

LA—left atrium, LAVmax—maximal left atrium volume, LAVmin—minimal left atrium volume, LAVpre—pre-atrial contraction left atrium volume.

**Table 5 medicina-58-01283-t005:** Low right atrial low right ventricular pacing group.

	Baseline	1 Month	3 Months	*p*-Value Baseline vs. 1 Month	*p*-Value Baseline vs. 3 Months
LA volumetric parameters					
LAVmax, mL	72.5 ± 27.2	77.8 ± 23.9	81.5 ± 21.4	0.245	0.286
LAVmax index, mL/m^2^	36.5 ± 11.9	39.5 ± 10.4	39.6 ± 9.1	0.187	0.213
LAVpre, ml	50.2 ± 20.2	56.2 ± 18.4	56.2 ± 18.4	0.191	0.505
LAVpre index, mL/m^2^	25.2 ± 8.8	28.3 ± 8.7	27.2 ± 8.1	0.191	0.477
LAVmin, mL	35.6 ± 16.1	38.2 ± 13.5	39.2 ± 12.6	0.408	0.594
LAVmin index, mL/m^2^	17.9 ± 7.3	19.4 ± 6.4	19.0 ± 5.5	0.301	0.625
LA functional parameters					
Total emptying fraction, %	51.6 ± 7.2	50.6 ± 7.6	52.2 ± 7.8	0.660	0.534
Passive emptying fraction, %	31.1 ± 7.6	30.1 ± 8.5	31.8 ± 9.5	0.460	0.824
Active emptying fraction, %	29.7 ± 8.6	31.7 ± 8.7	29.8 ± 8.0	0.334	0.790
LA strain parameters					
Reservoir strain	23.7 ± 9.5	23.7 ± 9.1	24.2 ± 10.2	0.925	0.575
Conduit strain	−11.0 ± 3.7	−12.4 ± 7.4	−13.8 ± 8.0	0.778	0.161
Contractile strain	−12.7 ± 7.6	−11.2 ± 7.2	−10.0 ± 4.8	0.683	0.093
Stiffness	0.45 ± 0.32	0.51 ± 0.39	0.39 ± 0.18	0.518	0.334

LA—left atrium, LAVmax—maximal left atrium volume, LAVmin—minimal left atrium volume, LAVpre—pre-atrial contraction left atrium volume.

## Data Availability

Not applicable.

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
