# Peer review of "Atrial Pacing Affects Left Atrial Morphological and Functional Parameters Early after Pacemaker Implantation"

_medicina, 2022, doi:10.3390/medicina58091283_

Round 1

Reviewer 1 Report

Thank you very much for the opportunity to review an interesting manuscript: Atrial pacing affects left atrial morphological and functional parameters early after pacemaker implantation.

The value of the study is that it is a prospective study.

The second value is that the research concerns patients with DDD stimulation - currently the most commonly used. The number of marked echocardiographic parameters of the activity of the left atrium is impressive. Patients divided into three groups. The conclusions drawn from the work are logical. I suggest planning the next examination with the division of patients according to the place of stimulation of the right ventricle-is it a apex or interventricular septum. These elements can also affect the echocardiographic parameters of the left atrium.

The authors in the study do not specify where the electrode is implanted in the right ventricle.

In my opinion, this should be written in the limitations of the study. The place of implantation of the ventricular electrode affects the function of the left atrium. I agree that the authors in the study focused on echocardiographic parameters. However, the place of implantation of the ventricular electrode is important. The authors refer to the problem of the place of stimulation in the last sentences of the Discussion. However, this issue can be developed and the problem of ventricular stimulation can be addressed in the discussion. In the next tests, echocardiographic diagnostics can be extended to include the study of left ventricular function in various modes of stimulation.  Please cite the following article:  doi: 10.1161/CIRCEP.114.001360. PMID: 25336367.

Reviewer 2 Report

Authors mentioned the association between left atrial function and atrial pacing. The theme is interesting, but it seems to be hard to understand what authors intended to analyze and find.

The primary endpoint is obscure. I cannot understand why authors divided candidates as three groups. In addition, table 1 and 2 showed new four categories of “high / low atrial pacing with low / high RV pacing group”. It makes confusing for readers. Authors should demonstrate the protocol of the present study.

Comparison is too much. It may cause the issue of multiple comparison. I strongly recommend authors to confine the current analysis to essentials.

How did authors deal with the difference of categorical variables? In table 2, I could not find the statistical significance for “chronic renal failure” using Chi-square test (P=0.25 in my calculation). The P value for “diabetes mellitus” can also be different from the result of the test, although it had the statistical significance (P=0.043). Regarding other analysis of categorical analysis in table 1, it was apart from the result of chi-square test. If authors performed another analysis but for chi-square test, authors should mention it in method section. Otherwise, authors should make sure the result was correct by calculating all analysis again.

It would be true that patients with high-atrial pacing and low ventricular pacing tended to progress left atrial remodeling. However, I suppose such patients had already comprised the left atrial remodeling in basic period, because the left atrial size in the group was larger than the parameter of the other group. If noting the change from basic value, there were little differences between groups. For example, the change in LAVmin between baseline and 3 months, it was nearly the same, with 1.11 (42.7/38.2) in high-atrial burden with low RV pacing, and 1.11 (39.2/35.6) in low-atrial with low RV pacing. Although I cannot understand why the difference occurred, I doubt it would be originated from limited number of candidates or missing patients in low-atrial with low RV pacing. I cannot trust the authors’ result.

In the current population, patients with high-atrial and low ventricular pacing were patients with SND. On the other hand, patients with low-atrial and low ventricular pacing were those with AVB40. How did authors manage the etiological difference? I suppose patients who had low RV pacing, instead of AVB would be healthier than those with normal AVB.

Although authors demonstrated too much information of the review of previous literatures in discussion section, there was few the sentences to contemplate why authors obtained the result. I feel it uncomfortable.
